# IWBVT: Instance Weighting-based Bias-Variance Trade-off for Crowdsourcing

**Wenjun Zhang**
School of Computer Science
China University of Geosciences
Wuhan 430074, China
wjzhang@cug.edu.cn

**Liangxiao Jiang**[*]
School of Computer Science
China University of Geosciences
Wuhan 430074, China
ljiang@cug.edu.cn

**Chaoqun Li**
School of Mathematics and Physics
China University of Geosciences
Wuhan 430074, China
chqli@cug.edu.cn

## Abstract

In recent years, a large number of algorithms for label integration and noise correction have been proposed to infer the unknown true labels of instances in crowdsourcing. They have made great advances in improving the label quality of crowdsourced datasets. However, due to the presence of intractable instances, these algorithms are usually not as significant in improving the model quality as they are in improving the label quality. To improve the model quality, this paper proposes an instance weighting-based bias-variance trade-off (IWBVT) approach. IWBVT at first proposes a novel instance weighting method based on the complementary set and entropy, which mitigates the impact of intractable instances and thus makes the bias and variance of trained models closer to the unknown true results. Then, IWBVT performs probabilistic loss regressions based on the bias-variance decomposition, which achieves the bias-variance trade-off and thus reduces the generalization error of trained models. Experimental results indicate that IWBVT can serve as a universal post-processing approach to significantly improving the model quality of existing state-of-the-art label integration algorithms and noise correction algorithms. Our codes and datasets are available at https://github.com/jiangliangxiao/IWBVT.

## 1   Introduction

Crowdsourcing eases the difficulty of obtaining training datasets for supervised learning [8]. In crowdsourcing scenarios, instances are annotated not by domain experts but by crowd workers from crowdsourcing platforms [1, 12]. While crowd workers are more cost-effective compared to domain experts, they typically possess inferior expertise and are thus more prone to assigning noisy labels [2, 13]. To mitigate the impact of noisy labels, a common practice in crowdsourcing is *repeated annotating*, where each instance is annotated by multiple workers to obtain multiple noisy labels [21]. Subsequently, a multitude of algorithms have been proposed to infer the unknown true label of an instance from its multiple noisy labels [3, 11, 15].

---

[*]Corresponding author

38th Conference on Neural Information Processing Systems (NeurIPS 2024).

Specifically, these proposed algorithms can be roughly classified into two categories, namely *label integration* algorithms and *noise correction* algorithms. Label integration algorithms aim to integrate multiple noisy labels of each instance to infer an integrated label that is as close as possible to its unknown true label [16, 30]. Noise correction algorithms focus on identifying and correcting noise in integrated labels obtained from label integration algorithms [19, 32]. Therefore, both label integration algorithms and noise correction algorithms inevitably pay more attention to the label quality of crowdsourced datasets, i.e., the proportion of instances in crowdsourced datasets whose integrated labels are equal to the unknown true labels. Indeed, these proposed algorithms have achieved empirical success in improving the label quality of crowdsourced datasets.

However, due to the presence of intractable instances, these algorithms often fall short of achieving anticipated improvements in the model quality. Here, the model quality is the proportion of instances whose predicted labels are equal to the unknown true labels when models classify test instances. On the one hand, the proportion of intractable instances in datasets tends to be low, which makes them have less impact on the label quality. For this reason, the above algorithms pay little attention to intractable instances in improving the label quality. On the other hand, the reason why intractable instances are hard to label and infer is that their attributes are ambiguous. Ambiguous attributes affect the effectiveness of models in learning classification rules from crowdsourced datasets. Therefore, intractable instances have a greater impact on the model quality compared to the label quality. Considering the fact that collecting high-quality integrated labels is ultimately aimed at training high-quality models, improving the model quality should be paid more attention compared to improving the label quality in crowdsourcing.

To improve the model quality, this paper proposes an instance weighting-based bias-variance trade-off (IWBVT) approach for crowdsourcing. IWBVT at first proposes a novel instance weighting method based on the idea of complementary set and entropy, which mitigates the impact of intractable instances and thus makes the bias and variance of trained models closer to the unknown true results. Subsequently, IWBVT performs probabilistic loss regressions based on the bias-variance decomposition, which achieves the bias-variance trade-off and thus reduces the generalization error of trained models. In general, the contributions of this paper can be summarized as follows:

- We focus on the performance of existing algorithms in terms of the model quality and reveal that existing algorithms are not as significant in improving the model quality as they are in improving the label quality.

- We propose a novel instance weighting method based on the complementary set and entropy. This new instance weighting method is more robust and can be applied to more complex crowdsourced scenarios.

- We propose IWBVT to improve the model quality. IWBVT mitigates the impact of intractable instances by instance weighting and achieves the bias-variance trade-off by probabilistic loss regressions.

- We demonstrate that IWBVT can serve as a universal post-processing approach to significantly improving the model quality of existing state-of-the-art label integration algorithms and noise correction algorithms.

## 2 Related work

With *repeated annotating*, crowdsourcing collects multiple noisy labels for each instance in datasets. Subsequently, label integration is usually used to integrate multiple noisy labels to infer the unknown true label for each instance. Initiating the area of label integration, [4] leveraged an expectation-maximization (EM) algorithm to estimate a confusion matrix, which models workers and class priors in clinical diagnostics. In contrast, [20] performed majority voting based on multiple noisy labels of instances, and the class receiving the highest number of votes was determined as the integrated label. Enhancing this concept by incorporating worker reliability, [10] performed weighted majority voting by iteratively estimating worker weights and integrated labels. [23] further proposed max-margin majority voting, which integrates labels by maximizing the margin between classes. Recently, [3] augmented the label space for an instance by considering the labels of its neighbors, which distinguishes the impact of different neighbors by instance weighting. Inspired by label distribution learning [6, 17, 26], [8] proposed multiple noisy label distribution propagation, which absorbs label distributions of neighboring instances into the label distribution of the focal instance.

No matter how powerful the label integration algorithms are, a certain degree of noise is always present in integrated labels. Subsequently, noise correction has been proposed to identify and correct these noises. Initiating the area of noise correction, [19] introduced three distinct algorithms: *polishing labels* (PL), *self-training correction* (STC), and *cluster-based correction* (CC). PL divides datasets into subsets to train multiple models and then performs majority voting based on the models' predictions to correct the original integrated labels. STC filters the dataset into a clean set and a noise set, iteratively training models on the clean set to predict and correct instances in the noise set. CC estimates the probability of each instance belonging to each class through repeated clustering and performs weighted majority voting to correct original integrated labels. Forgoing the above three algorithms, [32] adaptively estimated the proportion of noise in datasets based on multiple noisy labels to filter out a clean set and a noise set. Recently, drawing from multi-view learning [31], [15] used multi-view learning for correcting noise instances. They trained dual models on the attribute and multiple noisy label views of the clean set to correct instances in the noise set. [11] focused on the effect of neighboring instances on noise filtering before noise correction, utilizing multiple noisy label distributions of neighbors to identify noise instances more accurately.

In essence, both label integration algorithms and noise correction algorithms aim to improve the model quality by improving the label quality. Unfortunately, due to the presence of intractable instances, these algorithms are usually not as significant in improving the model quality as they are in improving the label quality. Currently, although there exist several supervised or semi-supervised approaches focused on improving the model quality from noisy training datasets [7, 14, 28], they are not perfectly applicable to crowdsourcing. On the one hand, they cannot utilize the multiple noisy labels specific to crowdsourcing. On the other hand, semi-supervised approaches typically assume that true labels of a few instances are known, which cannot be satisfied in crowdsourcing. In this context, we propose IWBVT, as the first universal post-processing approach to improve the model quality of both label integration algorithms and noise correction algorithms in crowdsourcing.

## 3    Notations and preliminaries

Let $D$ denote a crowdsourced dataset $\{(\boldsymbol{x}_i, \boldsymbol{L}_i)\}_{i=1}^{N}$, where $N$ represents the number of instances, and $\boldsymbol{x}_i$ is the $i$-th instance, represented as $\{x_{i1}, \ldots, x_{im}, \ldots, x_{iM}\}$. Here, $M$ signifies the dimension of attributes and $x_{im}$ denotes the attribute value of $\boldsymbol{x}_i$ on the $m$-th attribute $A_m$. $\boldsymbol{L}_i$ denotes multiple noisy labels of $\boldsymbol{x}_i$, which can be represented as $\{l_{ir}\}_{r=1}^{R}$. $R$ denotes the number of workers, $l_{ir}$ denotes the label of $\boldsymbol{x}_i$ annotated by the $r$-th worker $u_r$. $l_{ir}$ takes a value from $\{-1, c_1, \ldots, c_q, \ldots, c_Q\}$, where $Q$ denotes the number of classes, $c_q$ denotes the $q$-th class and $-1$ denotes that $u_r$ does not annotate $\boldsymbol{x}_i$. The purpose of label integration and noise correction is to infer an integrated label $\hat{y}_i$ for $\boldsymbol{x}_i$ and to minimize the error between $\hat{y}_i$ and the unknown true label $y_i$.

### 3.1    Instance weighting for crowdsourcing

Given $(\boldsymbol{x}_i, \boldsymbol{L}_i)$, the weight of $\boldsymbol{x}_i$ is denoted by $w_i$. Intuitively, the smaller the value of $w_i$, the more likely that $\boldsymbol{x}_i$ is an intractable instance. To estimate $w_i$, $\boldsymbol{L}_i$ is first transformed into a multiple noisy label distribution $\boldsymbol{P}_i = \{P(c_q|\boldsymbol{L}_i)\}_{q=1}^{Q}$, where the probability $P(c_q|\boldsymbol{L}_i)$ reflects the proportion of labels in $\boldsymbol{L}_i$ that take the value $c_q$. Subsequently, several representative instance weighting methods have been proposed based on $\boldsymbol{P}_i$. First, [21] proposed estimating $w_i$ by $P(\hat{y}_i|\boldsymbol{L}_i)$, i.e., $w_i \propto P(\hat{y}_i|\boldsymbol{L}_i)$. Take MV as an example, the probability $P(\hat{y}_i|\boldsymbol{L}_i)$ consistently equals the maximum value in $\boldsymbol{P}_i$. This method is usually effective when $Q = 2$. However, when $Q > 2$, $P(\hat{y}_i|\boldsymbol{L}_i)$ is not sufficient to distinguish different distributions, such as $\{0.5, 0.3, 0.2\}$ and $\{0.5, 0.4, 0.1\}$.

Subsequently, [27] proposed estimating $w_i$ by the entropy of $\boldsymbol{P}_i$, i.e., $w_i \propto \frac{1}{Ent(\boldsymbol{P}_i)}$, where

$$Ent(\boldsymbol{P}_i) = -\sum_{q=1}^{Q} P(c_q|\boldsymbol{L}_i) \log P(c_q|\boldsymbol{L}_i). \tag{1}$$

Based on the maximum entropy principle, when $\boldsymbol{P}_i$ conforms more closely to the uniform distribution, the entropy $Ent(\boldsymbol{P}_i)$ increases, leading to a decrease in the corresponding weight $w_i$. Though entropy-based methods can weight instances in multi-class datasets, they still fail to distinguish some complex distributions, such as $\{0.4, 0.3, 0.3\}$ and $\{0.4, 0.4, 0.2\}$. Recently, [3] proposed estimating $w_i$ by the

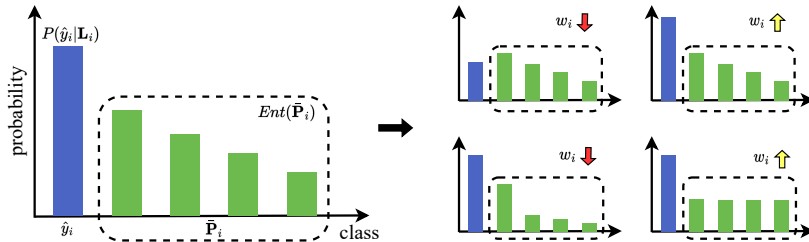

Figure 1: The illustration of our new instance weighting method.

class margin as follows:

$$w_i \propto \max(\boldsymbol{P}_i) - \sec(\boldsymbol{P}_i), \tag{2}$$

where $\max(\boldsymbol{P}_i)$ and $\sec(\boldsymbol{P}_i)$ denote the largest and second largest values in $\boldsymbol{P}_i$, respectively. This method focuses on the confusing classes in crowdsourced datasets. Nevertheless, it still struggles to distinguish some complex distributions, such as $\{0.5, 0.3, 0.1, 0.1\}$ and $\{0.4, 0.2, 0.2, 0.2\}$.

In addition to these methods, there are also a few methods that estimate $w_i$ with classification models or evolutionary algorithms [22, 29]. However, these methods have not been discussed here as their performance is affected by the selected models, loss functions, parameter settings, etc.

### 3.2 Bias-variance decomposition

The bias-variance decomposition is an effective way to analyze the generalization error of models. Referring to [9], given a model $f$, its generalization error $\mathbb{E}_f$ can be denoted as follows:

$$\mathbb{E}_f = \sum_{i=1}^{N} P(\boldsymbol{x}_i) \left( bias_i^2 + var_i + \sigma_i^2 \right), \tag{3}$$

where $P(\boldsymbol{x}_i)$ denotes the probability of selecting $\boldsymbol{x}_i$ from $D$. $bias_i^2$ and $var_i$ denote the bias term and the variance term, respectively. They are estimated as follows:

$$bias_i^2 = \frac{1}{2} \sum_{q=1}^{Q} \left[ P(c_q|\boldsymbol{x}_i) - P(c_q|f, \boldsymbol{x}_i) \right]^2, \tag{4}$$

$$var_i = \frac{1}{2} \left[ 1 - \sum_{q=1}^{Q} P(c_q|f, \boldsymbol{x}_i)^2 \right], \tag{5}$$

where $P(c_q|\boldsymbol{x}_i)$ denotes the true probability that $\boldsymbol{x}_i$ belongs to $c_q$, $P(c_q|f, \boldsymbol{x}_i)$ denotes the probability that $f$ classifies $\boldsymbol{x}_i$ into $c_q$ in multiple results generated by cross-validation. Therefore, $P(c_q|\boldsymbol{x}_i)$ is independent of $f$ and is only related to $D$. $\sigma_i^2$ denotes the noise term, which is also only related to $D$.

## 4 Approach

The primary objective of IWBVT is to improve the model quality through bias-variance trade-off. However, in crowdsourcing scenarios, $P(c_q|\boldsymbol{x}_i)$ can only be roughly estimated because $y_i$ is unknown and $\hat{y}_i$ is inaccurate. To estimate the bias and variance of $f$ as accurately as possible, IWBVT first mitigates the impact of intractable instances by instance weighting. Subsequently, to achieve the bias-variance trade-off, IWBVT learns probabilistic loss by probabilistic loss regressions.

### 4.1 Instance weighting

As previously mentioned, existing instance weighting methods struggle to distinguish certain complex distributions effectively. In response, IWBVT introduces a novel instance weighting method that leverages the complementary set and entropy to overcome this limitation. The innovative aspects of this method are depicted in Figure 1. As shown in the figure, $P(\hat{y}_i|\boldsymbol{L}_i)$ reflects the proportion

Table 1: The comparison results of instance weighting methods on complex distributions.

| Complex distributions | $P(\hat{y}_i\|\boldsymbol{L}_i)$ | $\frac{1}{Ent(\boldsymbol{P}_i)}$ | $\max(\boldsymbol{P}_i) - \sec(\boldsymbol{P}_i)$ | $P(\hat{y}_i\|\boldsymbol{L}_i)\frac{Ent(\bar{\boldsymbol{P}}_i)}{\log(Q-1)}$ |
|---|---|---|---|---|
| {0.5, 0.3, 0.2} & {0.5, 0.4, 0.1} | 0.50 & 0.50 ✗ | 0.67 & 0.73 ✗ | 0.20 & 0.10 ✓ | 0.70 & 0.52 ✓ |
| {0.4, 0.3, 0.3} & {0.4, 0.4, 0.2} | 0.40 & 0.40 ✗ | 0.64 & 0.66 ✗ | 0.10 & 0.00 ✓ | 0.58 & 0.53 ✓ |
| {0.5, 0.3, 0.1, 0.1} & {0.4, 0.2, 0.2, 0.2} | 0.50 & 0.40 ✓ | 0.59 & 0.52 ✓ | 0.20 & 0.20 ✗ | 0.62 & 0.58 ✓ |

of the integrated label in $\boldsymbol{L}_i$. A higher $P(\hat{y}_i|\boldsymbol{L}_i)$ suggests more workers reach a consensus on $\boldsymbol{x}_i$, thereby indicating a decreased likelihood of $\boldsymbol{x}_i$ being an intractable instance. To extend our method to multi-class datasets, we also focus on the entropy of $\bar{\boldsymbol{P}}_i$, i.e., $Ent(\bar{\boldsymbol{P}}_i)$. Here, $\bar{\boldsymbol{P}}_i$ is the complementary set of $\{P(\hat{y}_i|\boldsymbol{L}_i)\}$ in $\boldsymbol{P}_i$ ($\{P(c_q|\boldsymbol{L}_i)\}_{q=1}^Q$). Intuitively, workers tend to reach a consensus on a special class on tractable instances, so they should be more randomized on other classes. The entropy of $\bar{\boldsymbol{P}}_i$ reflects the degree of randomization. Accordingly, our instance weighting method considers four cases, shown to the right side of the arrow in Figure 1. Among them, the case in the upper left corner indicates that when $Ent(\bar{\boldsymbol{P}}_i)$ is fixed, a lower $P(\hat{y}_i|\boldsymbol{L}_i)$ results in a lower $w_i$. Conversely, the case in the upper right corner indicates that a higher $P(\hat{y}_i|\boldsymbol{L}_i)$ results in a higher $w_i$. The case in the lower left corner indicates that when $P(\hat{y}_i|\boldsymbol{L}_i)$ is fixed, a lower $Ent(\bar{\boldsymbol{P}}_i)$ results in a lower $w_i$. The case in the lower right corner indicates that the higher $Ent(\bar{\boldsymbol{P}}_i)$ results in a higher $w_i$. To cover these cases, specifically, we estimate $w_i$ as follows:

$$w_i = P(\hat{y}_i|\boldsymbol{L}_i)\frac{Ent(\bar{\boldsymbol{P}}_i)}{\log(Q-1)}, \tag{6}$$

where $\log(Q-1)$ is the normalization factor. When $Q = 2$, we set $\frac{Ent(\bar{\boldsymbol{P}}_i)}{\log(Q-1)}$ to 1.

To demonstrate the superiority of our weighting method over existing methods, we calculate instance weights with each method on all the complex distributions mentioned in Section 3.1. Table 1 reports the detailed comparison results. Here, "✓" and "✗" indicate whether the weighting method is effective in distinguishing the corresponding complex distribution, respectively. Empirically, the weight corresponding to the front distribution in each example should be higher than the latter. The results show that only our method can distinguish all these types of complex distributions, while existing methods cannot. By Eq. (6), we calculate weights of all instances as $\boldsymbol{W} = \{w_i\}_{i=1}^N$.

**Theorem 1.** *When $Ent(\bar{\boldsymbol{P}}_i)$ remains constant, Eq. (6) covers $w_i \propto P(\hat{y}_i|\boldsymbol{L}_i)$. When $Q > 2$ and $P(\hat{y}_i|\boldsymbol{L}_i)$ is the maximum value in $\boldsymbol{P}_i$, Eq. (6) covers $w_i \propto \max(\boldsymbol{P}_i) - \sec(\boldsymbol{P}_i)$.*

*Proof.* When $Q = 2$, we set $Ent(\bar{\boldsymbol{P}}_i)/\log(Q-1)$ to 1, so Eq. (6) simplifies to $w_i \propto P(\hat{y}_i|\boldsymbol{L}_i)$. $w_i \propto P(\hat{y}_i|\boldsymbol{L}_i)$ still holds in Eq. (6) when $Q > 2$ and $Ent(\bar{\boldsymbol{P}}_i)$ remains constant. When $Q > 2$ and $P(\hat{y}_i|\boldsymbol{L}_i)$ remains constant, $w_i \propto Ent(\bar{\boldsymbol{P}}_i)$ holds. According to the maximum entropy principle, $Ent(\bar{\boldsymbol{P}}_i)$ takes its maximum value when any element of $\bar{\boldsymbol{P}}_i$ is equal to $\frac{1-P(\hat{y}_i|\boldsymbol{L}_i)}{Q-1}$. At this point, if $P(\hat{y}_i|\boldsymbol{L}_i)$ is the maximum value in $\boldsymbol{P}_i$, $\max(\boldsymbol{P}_i) - \sec(\boldsymbol{P}_i)$ takes its maximum value. Conversely, when $Ent(\bar{\boldsymbol{P}}_i)$ takes its minimum value, $\max(\boldsymbol{P}_i) - \sec(\boldsymbol{P}_i)$ also takes its minimum value. Therefore, $w_i \propto \max(\boldsymbol{P}_i) - \sec(\boldsymbol{P}_i)$ holds when $Q > 2$ and $P(\hat{y}_i|\boldsymbol{L}_i)$ is the maximum value in $\boldsymbol{P}_i$. Due to the limited pages, more detailed proof of Theorem 1 is provided in Appendix A. □

### 4.2 Bias-variance trade-off

After instance weighting, the impact of intractable instances is mitigated. Therefore, $P(c_q|\boldsymbol{x}_i)$ can be calculated more accurately, and then bias and variance can be estimated more accurately. Based on this result, the bias-variance trade-off will be more effective. After instance weighting, we train a classification model $f$ on $D$ with $\boldsymbol{W}$. Let $C$ denote the label space $\{c_1, \ldots, c_q, \ldots, c_Q\}$, in this paper, $f$ classifies the test instance $\boldsymbol{x}$ with the bias-variance trade-off as follows:

$$c(\boldsymbol{x}) = \underset{c_q \in C}{\arg\max}(f(c_q|\boldsymbol{x}) + h_q(\boldsymbol{x})), \tag{7}$$

where $f(c_q|\boldsymbol{x})$ denotes the probability that $\boldsymbol{x}$ belongs to $c_q$ predicted by $f$. $h_q(\boldsymbol{x})$ is the prediction of the regression model $h_q$ trained on the following probabilistic loss regression task:

$$T_q = (\boldsymbol{\mathcal{X}}; \boldsymbol{W}; \boldsymbol{\mathcal{L}}_q), \tag{8}$$

where $\mathcal{X}$ is the attribute matrix consisting of all training instances. $\mathcal{L}_q$ is the probabilistic loss vector for $c_q$, which can be represented as $\{\mathcal{L}_{1q}, \ldots, \mathcal{L}_{iq}, \ldots, \mathcal{L}_{Nq}\}^T$. $\mathcal{L}_{iq}$ is calculated as follows:

$$\mathcal{L}_{iq} = \begin{cases} 1 - f(c_q|\boldsymbol{x}_i) & c_q = \hat{y}_i \\ 0 - f(c_q|\boldsymbol{x}_i) & c_q \neq \hat{y}_i \end{cases}. \tag{9}$$

**Theorem 2.** *When the probabilistic loss is defined as in Eq. (9), performing probabilistic loss regressions constructed by Eq. (8) ensures that Eq. (7) asymptotically achieves the bias-variance trade-off.*

*Proof.* When $f$ is adjusted, $P(c_q|f, \boldsymbol{x}_i)$ changes with $f$, and this change is denoted as $\Delta_{iq}$. Let $\tilde{bias}_i^2$ and $\tilde{var}_i$ denote the changed bias term and variance term, they can be calculated as follows:

$$\tilde{bias}_i^2 = bias_i^2 + \sum_{q=1}^{Q} \Delta_{iq} P(c_q|f, \boldsymbol{x}_i) + \frac{1}{2} \sum_{q=1}^{Q} \Delta_{iq}^2 - \sum_{q=1}^{Q} \Delta_{iq} P(c_q|\boldsymbol{x}_i). \tag{10}$$

$$\tilde{var}_i = var_i - \sum_{q=1}^{Q} \Delta_{iq} P(c_q|f, \boldsymbol{x}_i) - \frac{1}{2} \sum_{q=1}^{Q} \Delta_{iq}^2. \tag{11}$$

Due to the limited pages, more detailed derivation of Eqs. (10) - (11) is provided in Appendix B. Comparing Eq. (10) and Eq. (11) shows that the common terms $\sum_{q=1}^{Q} \Delta_{iq} P(c_q|f, \boldsymbol{x}_i)$ and $\frac{1}{2} \sum_{q=1}^{Q} \Delta_{iq}^2$ in $\tilde{bias}_i^2$ and $\tilde{var}_i$ have opposite signs. Therefore, when improving $\mathbb{E}_f$, the bias and variance tend to change in opposite trends, which is known as the bias-variance dilemma. Improving $\mathbb{E}_f$ by synergistically considering changes in both bias and variance is known as the bias-variance trade-off. According to Eqs. (3), (10), and (11), we can get the changed $\tilde{\mathbb{E}}_f$ as follows:

$$\tilde{\mathbb{E}}_f = \mathbb{E}_f - \sum_{i=1}^{N} P(\boldsymbol{x}_i) \sum_{q=1}^{Q} \Delta_{iq} P(c_q|\boldsymbol{x}_i). \tag{12}$$

In general, when $c_q$ is the true label of $\boldsymbol{x}_i$, $P(c_q|\boldsymbol{x}_i)$ tends to 1, otherwise it tends to 0. However, the true label $y_i$ is unknown in crowdsourcing scenarios. After instance weighting, the impact of intractable instances is mitigated, so we assume that $\hat{y}_i$ is equal to $y_i$. Therefore, when $c_q \neq \hat{y}_i$, $\Delta_{iq} P(c_q|\boldsymbol{x}_i)$ tends to 0. When $c_q = \hat{y}_i$, since the probability terms $P(\boldsymbol{x}_i)$ and $P(c_q|\boldsymbol{x}_i)$ in Eq. (12) are non-negative, $\tilde{\mathbb{E}}_f$ is guaranteed to be less than $\mathbb{E}_f$ as long as $\Delta_{iq}$ is greater than 0. In summary, the key factor of the bias-variance trade-off is $\Delta_{iq} (c_q = \hat{y}_i)$. To make $\Delta_{iq} (c_q = \hat{y}_i)$ greater than 0, the following optimization task can be constructed:

$$\begin{aligned} \underset{\boldsymbol{x}_i}{maximize} \quad & f(\hat{y}_i|\boldsymbol{x}_i) \\ s.t. \quad & f(\hat{y}_i|\boldsymbol{x}_i) - \max_{c_q \in C \wedge c_q \neq \hat{y}_i} f(c_q|\boldsymbol{x}_i) \geq 0. \end{aligned} \tag{13}$$

Here, maximizing $f(\hat{y}_i|\boldsymbol{x}_i)$ ensures that $\Delta_{iq} (c_q = \hat{y}_i)$ is greater than 0, while the constraint ensures that the prediction of $f$ will be $\hat{y}_i$. Then, according to the Lagrange multiplier, the Lagrange function can be constructed as follows:

$$L(\boldsymbol{x}_i) = f(\hat{y}_i|\boldsymbol{x}_i) + \lambda\big[f(\hat{y}_i|\boldsymbol{x}_i) - \max_{c_q \in C \wedge c_q \neq \hat{y}_i} f(c_q|\boldsymbol{x}_i)\big], \tag{14}$$

where $\lambda \geq 0$. For simplicity, $L'(\boldsymbol{x}_i)$ can be further constructed as follows:

$$\begin{aligned} L'(\boldsymbol{x}_i) &= L(\boldsymbol{x}_i) - \max_{c_q \in C \wedge c_q \neq \hat{y}_i} f(c_q|\boldsymbol{x}_i) \\ &= (1 + \lambda)\big[f(\hat{y}_i|\boldsymbol{x}_i) - \max_{c_q \in C \wedge c_q \neq \hat{y}_i} f(c_q|\boldsymbol{x}_i)\big]. \end{aligned} \tag{15}$$

Since the probability $\max_{c_q \in C \wedge c_q \neq \hat{y}_i} f(c_q|\boldsymbol{x}_i) \geq 0$, so $L(\boldsymbol{x}_i) \geq L'(\boldsymbol{x}_i)$. Ultimately, Eq. (13) can be optimized to achieve a better result by maximizing $L'(\boldsymbol{x}_i)$. At the same time, since $\lambda \geq 0$, the value of $L'(\boldsymbol{x}_i)$ is positively correlated with the following difference:

$$f(\hat{y}_i|\boldsymbol{x}_i) - \max_{c_q \in C \wedge c_q \neq \hat{y}_i} f(c_q|\boldsymbol{x}_i). \tag{16}$$

According to Eq. (9), through probabilistic loss regressions, when $c_q = \hat{y}_i$, $f(c_q|\boldsymbol{x}) + h_q(\boldsymbol{x})$ in Eq. (7) tends to 1. Conversely, when $c_q \neq \hat{y}_i$, $f(c_q|\boldsymbol{x}) + h_q(\boldsymbol{x})$ tends to 0. Therefore, Eq. (7) is effective in maximizing the difference Eq. (16). Ultimately, Theorem 2 is proved. $\square$

The whole learning process of IWBVT is shown in Algorithm 1. In Algorithm 1, lines 1-3 learn a weight for each instance and their time complexity is $O(NRQ)$. Line 4 trains a classification model $f$ whose training time complexity is denoted as $O(t_1)$. Lines 5-11 learn a probabilistic loss regression model $h_q$ for each class $c_q$ and their time complexity is $O(Q(Nt_2 + t_3))$. Here, $t_2$ denotes the prediction time complexity of $f$ on each class and $t_3$ denotes the training time complexity of $h_q$. In this paper, we select NB as the classification model and linear regression as the regression model. Therefore, $t_1$, $t_2$, $t_3$ are equal to $O(NM)$, $O(M)$, and $O(NM^2 + M^3)$, respectively. If only the highest order terms are taken, the time complexity of IWBVT is $O(NRQ + NQM^2 + QM^3)$.

---

**Algorithm 1** The learning process of IWBVT

---

**Require:** $\hat{D} = \{(\boldsymbol{x}_i, \boldsymbol{L}_i, \hat{y}_i)\}_{i=1}^N$ - a crowdsourced dataset with integrated labels.
**Ensure:** classification model $f$, regression model set $\boldsymbol{H}$.
 1: **for** $i = 1$ to $N$ **do**
 2:     Calculate the weight $w_i$ of $\boldsymbol{x}_i$ by Eq. (6);
 3: **end for**
 4: Train the classification model $f$ on $\hat{D}$ with $\boldsymbol{W} = \{w_i\}_{i=1}^N$;
 5: **for** $q = 1$ to $Q$ **do**
 6:     **for** $i = 1$ to $N$ **do**
 7:         Calculate the probabilistic loss $\mathcal{L}_{iq}$ by Eq. (9);
 8:     **end for**
 9:     Construct the regression task $T_q$ by Eq. (8);
10:     Learn the regression model $h_q$ on $T_q$;
11: **end for**
12: **return** classification model $f$, $\boldsymbol{H} = \{h_1, h_2, \ldots, h_Q\}$.

---

## 5  Experiments

To validate the effectiveness of IWBVT, we conduct a series of experiments on the whole 34 simulated and 2 real-world crowdsourced datasets published on the Crowd Environment and its Knowledge Analysis (CEKA) [33] platform. First, we illustrate the setup of our experiments, including comparison algorithms and their parameter settings. Next, we describe the simulation process and present the simulated experimental results in terms of the model quality. Finally, to further validate the strength of IWBVT, we analyze the experimental results of comparative experiments and ablation experiments on real-world crowdsourced datasets.

### 5.1  Experimental setup

We select seven state-of-the-art algorithms for experiments, including: *majority voting* (MV) [20], *iterative weighted majority voting* (IWMV) [10], *label augmented and weighted majority voting* (LAWMV) [3], *multiple noisy label distribution propagationg* (MNLDP) [8], *adaptive voting noise correction* (AVNC) [32], *multi-view-based noise correction* (MVNC) [15], and *neighborhood weighted voting-based noise correction* (NWVNC) [11]. Among them, MV is the simplest label integration algorithm and is used as a baseline for all algorithms. IWMV, LAWMV and MNLDP are three state-of-the-art label integration algorithms. AVNC, MVNC and NWVNC are three state-of-the-art noise correction algorithms. They are used to validate the effectiveness of IWBVT for label integration and noise correction. All these algorithms are implemented based on the CEKA platform and their parameter settings are consistent with the corresponding published papers. AVNC, MVNC and NWVNC are all performed based on integrated labels inferred by MV. Besides, we use linear regression as $h_q$ in IWBVT. All experiments are conducted on a Windows 10 machine with an AMD Athlon(tm) X4 860K Quad Core Processor @ 3.70 GHz and 16 GB of RAM.

### 5.2  Experiments on simulated datasets

**Datasets and simulation process.**    We conduct our simulated experiments on all simulated datasets published on the CEKA platform. These datasets come from a wide variety of application domains and represent plentiful crowdsourcing scenarios. Considering that the selected label integration

Table 2: The model quality (%) comparisons of MV, IWMV, LAWMV, MNLDP, AVNC, MVNC and NWVNC before and after using IWBVT on 34 simulated datasets.

| Dataset | MV | | IWMV | | LAWMV | | MNLDP | | AVNC | | MVNC | | NWVNC | |
|---|---|---|---|---|---|---|---|---|---|---|---|---|---|---|
| | ORI | IWBVT | ORI | IWBVT | ORI | IWBVT | ORI | IWBVT | ORI | IWBVT | ORI | IWBVT | ORI | IWBVT |
| anneal | 68.60 | 80.42 • | 68.27 | 81.29 • | 75.18 | 80.27 • | 67.84 | 81.15 • | 74.83 | 81.39 • | 73.94 | 81.96 • | 67.84 | 79.49 • |
| audiology | 59.93 | 61.39 | 58.05 | 60.08 | 61.15 | 63.49 | 58.43 | 60.50 | 68.93 | 65.49 ○ | 64.77 | 64.09 | 59.10 | 61.50 |
| autos | 54.31 | 59.98 • | 52.13 | 58.82 • | 56.20 | 60.73 • | 56.12 | 59.24 • | 57.55 | 59.87 | 57.10 | 60.61 • | 56.51 | 60.01 • |
| balance-scale | 86.57 | 87.67 | 88.38 | 88.68 | 88.62 | 89.02 | 88.95 | 89.24 | 84.98 | 87.37 • | 84.34 | 87.12 • | 86.58 | 88.28 • |
| biodeg | 68.34 | 72.28 • | 68.33 | 72.26 • | 76.05 | 77.02 | 71.32 | 75.33 • | 74.01 | 76.95 • | 70.43 | 75.19 • | 72.76 | 74.77 • |
| breast-cancer | 70.41 | 67.75 ○ | 70.24 | 68.13 | 72.50 | 71.56 | 70.55 | 69.90 | 72.86 | 72.80 | 72.89 | 72.22 | 72.62 | 72.83 |
| breast-w | 96.18 | 94.99 ○ | 96.12 | 95.10 ○ | 96.04 | 96.07 | 96.11 | 96.15 | 96.01 | 95.87 | 96.29 | 96.25 | 96.16 | 96.16 |
| car | 77.88 | 81.94 • | 80.09 | 83.69 • | 71.12 | 72.28 • | 73.07 | 76.36 • | 80.02 | 81.59 • | 76.93 | 81.25 • | 75.81 | 78.72 • |
| credit-a | 77.55 | 80.19 • | 77.91 | 79.88 • | 81.64 | 82.99 • | 78.36 | 80.81 • | 82.49 | 83.41 | 79.61 | 81.12 • | 80.07 | 81.72 • |
| credit-g | 72.38 | 72.54 | 72.55 | 73.04 | 73.29 | 72.89 | 72.72 | 72.97 | 74.12 | 74.31 | 73.47 | 73.96 | 73.76 | 74.41 |
| diabetes | 73.85 | 74.13 | 74.24 | 74.27 | 72.43 | 73.01 | 74.65 | 75.24 | 74.75 | 75.33 | 74.49 | 75.20 | 74.59 | 75.46 |
| heart-c | 82.86 | 81.45 | 82.65 | 81.95 | 84.05 | 84.09 | 83.72 | 83.08 | 83.10 | 82.52 | 83.34 | 83.23 | 83.67 | 83.37 |
| heart-h | 82.74 | 81.77 | 82.70 | 81.51 | 84.17 | 83.57 | 83.50 | 82.58 | 83.95 | 83.99 | 83.31 | 82.33 | 83.55 | 83.07 |
| heart-statlog | 82.52 | 79.89 ○ | 82.81 | 80.78 ○ | 84.78 | 84.19 | 83.81 | 82.56 | 83.52 | 82.56 | 83.33 | 82.37 | 84.26 | 84.00 |
| hepatitis | 78.98 | 76.97 | 79.72 | 76.50 ○ | 85.78 | 84.32 | 84.05 | 83.18 | 83.15 | 81.32 | 82.89 | 80.95 | 83.45 | 83.05 |
| horse-colic | 74.69 | 76.90 • | 75.12 | 76.63 | 76.84 | 80.69 • | 73.94 | 77.55 • | 80.67 | 82.35 | 77.29 | 80.59 • | 75.37 | 79.50 • |
| hypothyroid | 93.50 | 93.66 | 92.57 | 93.49 • | 92.29 | 92.29 | 93.68 | 93.76 | 95.07 | 95.14 | 94.44 | 94.40 | 93.84 | 93.99 |
| ionosphere | 81.23 | 82.22 | 81.08 | 81.57 | 77.90 | 81.39 • | 81.03 | 86.24 • | 83.05 | 86.46 • | 81.45 | 85.12 • | 79.32 | 87.03 • |
| iris | 90.67 | 93.07 • | 91.33 | 93.47 • | 95.53 | 95.60 | 95.13 | 95.40 | 95.53 | 95.47 | 94.73 | 95.73 | 95.80 | 95.80 |
| kr-vs-kp | 85.26 | 93.22 • | 85.29 | 93.19 • | 80.48 | 90.07 • | 81.27 | 91.14 • | 87.22 | 94.42 • | 85.57 | 93.90 • | 83.36 | 92.23 • |
| labor | 84.30 | 77.28 ○ | 79.90 | 74.03 ○ | 90.20 | 86.57 | 88.38 | 86.87 | 85.53 | 82.15 | 82.35 | 79.88 | 86.82 | 84.73 |
| letter | 63.48 | 64.55 • | 62.94 | 64.38 • | 63.95 | 64.77 | 64.00 | 64.73 | 64.05 | 65.09 • | 64.05 | 64.93 • | 63.58 | 64.63 • |
| lymph | 79.53 | 74.68 ○ | 80.45 | 75.40 ○ | 80.82 | 79.63 | 80.32 | 79.00 | 79.17 | 76.94 | 80.18 | 77.82 | 80.64 | 81.34 |
| mushroom | 91.72 | 97.07 • | 91.75 | 97.03 • | 89.23 | 89.37 | 93.42 | 97.15 • | 95.47 | 97.97 • | 95.06 | 97.84 • | 92.61 | 96.72 • |
| segment | 74.06 | 83.91 • | 72.41 | 82.59 • | 79.94 | 86.52 • | 79.78 | 86.40 • | 80.71 | 87.10 • | 80.09 | 86.49 • | 78.88 | 86.15 • |
| sick | 46.48 | 93.88 • | 46.61 | 93.93 • | 89.10 | 92.83 • | 75.59 | 93.94 • | 75.44 | 87.40 • | 63.52 | 93.92 • | 46.60 | 93.52 • |
| sonar | 66.18 | 63.26 | 66.23 | 63.49 | 66.33 | 68.01 | 65.99 | 69.35 • | 66.34 | 67.36 | 65.99 | 65.40 | 64.73 | 67.86 • |
| spambase | 71.84 | 76.11 • | 71.73 | 76.10 • | 74.03 | 81.33 • | 69.71 | 77.14 • | 74.96 | 79.25 • | 69.65 | 77.24 • | 71.13 | 73.13 • |
| tic-tac-toe | 68.78 | 68.86 | 68.54 | 69.11 | 70.63 | 71.00 | 65.59 | 65.90 | 71.02 | 71.02 | 70.09 | 70.55 | 71.02 | 71.52 |
| vehicle | 43.54 | 67.98 • | 44.47 | 68.28 • | 43.25 | 66.02 • | 42.85 | 66.95 • | 43.72 | 65.85 • | 43.99 | 68.44 • | 45.11 | 66.42 • |
| vote | 89.36 | 90.80 • | 89.35 | 90.48 | 89.20 | 89.63 | 89.32 | 90.43 | 90.29 | 93.18 • | 89.47 | 90.67 | 89.39 | 90.38 |
| vowel | 59.28 | 63.31 • | 59.51 | 62.92 • | 59.56 | 62.94 • | 63.01 | 64.32 | 60.24 | 62.65 • | 62.05 | 64.07 • | 60.35 | 63.21 • |
| waveform | 78.45 | 82.31 • | 77.73 | 81.40 • | 79.51 | 81.45 • | 79.57 | 81.70 • | 80.40 | 82.83 • | 78.74 | 82.43 • | 79.12 | 81.76 • |
| zoo | 89.73 | 90.48 | 89.30 | 90.38 | 90.25 | 91.98 | 90.90 | 91.77 | 88.27 | 88.81 | 87.63 | 87.66 | 89.92 | 91.70 |
| **Average** | 75.45 | 79.03 | 75.31 | 78.94 | 78.00 | 80.22 | 76.96 | 80.24 | 78.57 | 80.77 | 77.16 | 80.44 | 76.42 | 80.54 |
| **W/T/L** | - | 17/12/5 | - | 16/13/5 | - | 13/21/0 | - | 15/19/0 | - | 15/18/1 | - | 17/17/0 | - | 18/16/0 |

algorithms and noise correction algorithms handle the missing values of datasets differently, we use the unsupervised attribute filter *ReplaceMissingValues* in the Waikato Environment and Knowledge Analysis (WEKA) [25] platform to replace all missing values. Specifically, *ReplaceMissingValues* uses the mean of numerical attribute values or the modes of the nominal attribute values from the available data to replace missing values. Subsequently, to generate multiple noisy labels for each instance, we simulate the crowdsourcing process for these datasets. First, we randomly generate five workers whose label quality follows a normal distribution with N(0.65, $0.05^2$). The label quality of a worker reflects the probability that the noisy label annotated by this worker to an instance is the same as this instance's unknown true label. Then, we hide true labels and use these simulated workers to annotate datasets. Finally, we use the selected algorithms to infer integrated labels for these datasets. For each simulation, we evaluate the original model quality and the corresponding model quality improved using IWBVT through stratified 10-fold cross-validation. Here, we use Naive Bayes (NB) [5] as the target model. The above processes are repeated ten times independently for each algorithm on each dataset.

**Experimental results.** Table 2 shows the detailed model quality (%) comparisons of each algorithm on each dataset, respectively. The columns *ORI* and *IWBVT* correspond to the original model quality and the model quality using IWBVT, respectively. The symbols • and ○ in the table denote the model quality has a statistically significant improvement or degradation using our proposed IWBVT with a corrected paired two-tailed t-test with the significance level $\alpha = 0.05$ [18], respectively. Besides, the averages and the *Win/Tie/Lose* ($W/T/L$) values are summarized at the bottom of Table 2. The $W/T/L$ implies that when improving the original model quality, IWBVT wins on $W$ datasets, ties on $T$ datasets, and loses on $L$ datasets. These experimental results validate the effectiveness of IWBVT, and we can summarize the following highlights:

- The average model quality of MV using IWBVT on 34 datasets is 79.03%, which is higher than the original model quality of all selected algorithms. This demonstrates both the limitations of label integration algorithms or noise correction algorithms and the effectiveness of IWBVT in improving the model quality.

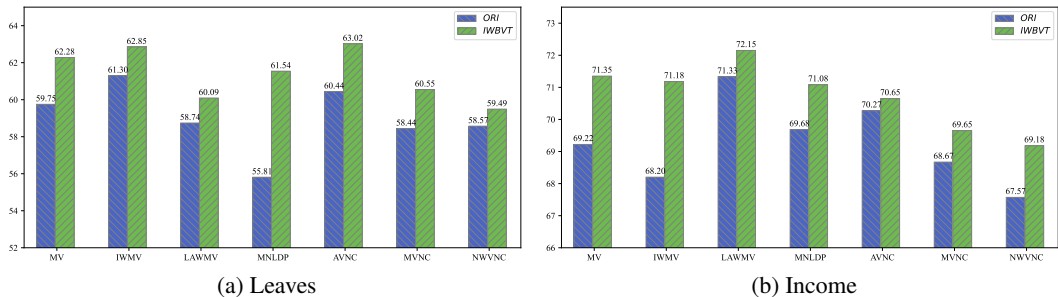

Figure 2: The model quality (%) comparisons of MV, IWMV, LAWMV, MNLDP, AVNC, MVNC and NWVNC before and after using IWBVT on *Leaves* and *Income*.

- The average model quality of IWMV (78.94%), LAWMV (80.22%), and MNLDP (80.24%) using IWBVT are higher than the original results of these state-of-the-art label integration algorithms. This demonstrates that IWBVT is still effective for more sophisticated label integration algorithms in improving the model quality.

- The average model quality of AVNC (80.77%), MVNC (80.44%), and NWVNC (80.54%) using IWBVT are also higher than the original results of these state-of-the-art noise correction algorithms. This demonstrates that IWBVT can serve as a universal post-processing approach to significantly improving the model quality.

- Based on the t-test results, the number of datasets in which IWBVT wins significantly ($W$) is always much higher than the number of datasets in which it loses significantly ($L$) for all algorithms. This strongly demonstrates the effectiveness and robustness of IWBVT.

### 5.3 Experiments on real-world datasets

**Datasets.** To demonstrate the robustness of IWBVT, the above simulation process pays more attention to common factors of crowdsourcing. However, training models on real-world datasets may also be affected by other factors, such as sparsity and annotating bias. To verify the effectiveness of IWBVT in real-world crowdsourced scenarios, we also construct our experiments on two widely used real-world crowdsourced datasets, *Leaves* and *Income*, published on the CEKA platform [34]. Here, *Leaves* and *Income* are selected through the online platform Amazon Mechanical Turk (AMT). *Leaves* is annotated by 83 workers and each instance is annotated by 10 workers. There are 6 classes, 384 instances, 3840 labels, 64 numeric attributes, and 0 missing values in *Leaves*. *Income* is annotated by 67 workers and each instance is also annotated by 10 workers. There are 2 classes, 600 instances, 6000 labels, 10 nominal attributes, and 0 missing values in *Income*. We only evaluate the original model quality and the corresponding model quality using IWBVT by stratified 10-fold cross-validation one time because real-world datasets do not have a random simulation process.

**Experimental results.** Figure 2 shows the model quality (%) comparisons of MV, IWMV, LAWMV, MNLDP, AVNC, MVNC and NWVNC before and after using IWBVT on *Leaves* and *Income*. With Figures 2a and 2b, we can find that IWBVT can also serve as a universal post-processing approach to significantly improving the model quality in real-world crowdsourced scenarios. Besides, we can also find the original model quality of several state-of-the-art algorithms is even lower than the original model quality of MV. These results once again demonstrate the limitation of label integration and noise correction in improving the model quality.

**Ablation experiment.** The above results only demonstrate the effectiveness of IWBVT as a whole, yet they do not delineate the contributions of its two key components: instance weighting and bias-variance trade-off. In IWBVT, instance weighting is used to mitigate the impact of intractable instances to make the bias and variance of trained models closer to the unknown true results. Therefore, to independently verify the effectiveness of instance weighting, we first observe the bias and variance of trained models before and after instance weighting. To estimate the bias and variance, referring to [24], NB is tested on *Leaves* and *Income* by ten runs of three-fold cross-validation. Figure 3a shows the bias and variance comparisons before and after using instance weighting on *Leaves* and *Income*.

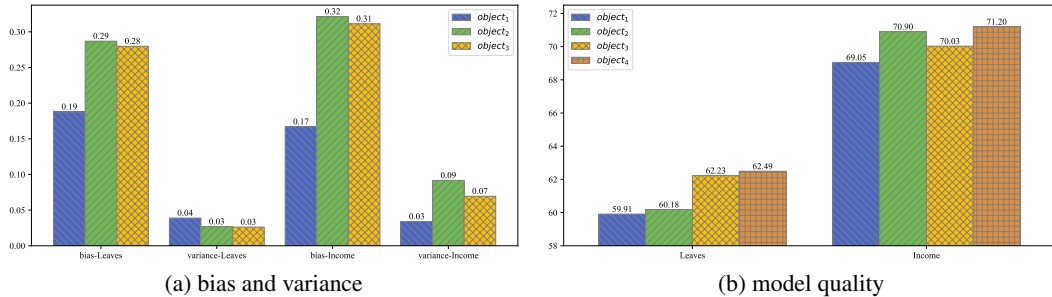

<table>
<tr><td>(a) bias and variance</td><td>(b) model quality</td></tr>
</table>

Figure 3: The ablation experiment comparisons of IWBVT and its components on *Leaves* and *Income*.

In Figure 3a, $object_1$ denotes the estimation results of NB trained directly with true labels. $object_2$ denotes the estimation results of NB trained with integrated labels inferred by MV. The conditions for $object_3$ and $object_2$ are the same, besides considering instance weighting. As can be seen in Figure 3a, the bias and variance of models trained only with integrated labels are usually higher than unknown true results. When instance weighting is introduced, both the bias and variance of models tend to be closer to unknown true results. These results demonstrate that instance weighting successfully corrects the bias and variance of trained models by mitigating the impact of intractable instances. Therefore, IWBVT performs the instance weighting before the bias-variance trade-off, which is more effective in improving the generalization performance of trained models.

Additionally, we also analyze the effectiveness of another component, the bias-variance trade-off, in improving model quality. Similarly, we still fix the label integration algorithm to be MV, and then introduce the instance weighting and bias-variance trade-off individually to observe their influence in improving model quality. Figure 3b shows the model quality (%) comparisons of MV using IWBVT or its components on *Leaves* and *Income*. In Figure 3b, $object_1$ denotes the model quality of MV, $object_2$ denotes the model quality of MV using the bias-variance trade-off, $object_3$ denotes the model quality of MV using the instance weighting, and $object_4$ denotes the model quality of MV using the whole IWBVT. As can be seen in Figure 3b, both the instance weighting and the bias-variance trade-off effectively improve the model quality of MV. This demonstrates that the two components of the IWBVT are both effective. Moreover, the model quality of MV using the whole IWBVT is the highest on both *Leaves* and *Income*, which suggests that it is reasonable for IWBVT to utilize both components at the same time. In addition, in Figure 3b, we can also find that the instance weighting is more effective on *Leaves*, while the bias-variance trade-off is more effective on *Income*. This is because the average label quality of *Leaves* is low. Instance weighting helps to identify rare instances that are inferred correctly, and therefore has a greater impact on *Leaves*. However, the average label quality of *Income* is high, and more instances can be correctly inferred than in *Leaves*. Therefore, the bias and variance are estimated closer to the unknown true values, so the bias-variance trade-off is more effective on *Income*.

## 6 Conclusion and future work

To improve the model quality of models trained on crowdsourced datasets, we propose a universal post-processing approach called IWBVT. IWBVT first mitigates the impact of intractable instances by instance weighting to make the bias and variance of trained models closer to the unknown true results. Then, IWBVT reduces the generalization error of trained models by the bias-variance trade-off. Experimental results suggest that IWBVT can significantly improve the model quality of existing state-of-the-art label integration algorithms and noise correction algorithms.

Though the above experimental results sufficiently demonstrate the effectiveness of IWBVT, some anomalies are found in experiments. Table 2 shows that IWBVT degrades the model quality on a few datasets such as *labor* and *lymph*. The datasets such as *labor* and *lymph* contain some numerical attributes that are significantly higher in magnitude than other attributes. However, the linear regression chosen for IWBVT in experiments is not robust to regression tasks constructed for these datasets. Therefore, in the future, we will further improve the robustness of IWBVT to make trained models insensitive to anomalous attributes.

## Acknowledgment

The work was partially supported by National Natural Science Foundation of China (62276241), Foundation of Key Laboratory of Artificial Intelligence, Ministry of Education, P.R. China (AI2022004), and Science and Technology Project of Hubei Province-Unveiling System (2021BEC007).

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

## Appendix A  Additions to the proof of Theorem 1

In the proof of **Theorem** 1, according to the maximum entropy principle, we get that $Ent(\bar{\boldsymbol{P}}_i)$ takes its maximum value when any element of $\bar{\boldsymbol{P}}_i$ is equal to $\frac{1-P(\hat{y}_i|\boldsymbol{L}_i)}{Q-1}$. Here, we provide its detailed derivation. First, by the definition of $\bar{\boldsymbol{P}}_i$, we simplify it to $\bar{\boldsymbol{P}}_i = \{P_q\}_{q=1}^{Q-1}$ with $\sum_{q=1}^{Q-1} P_q = 1 - P(\hat{y}_i|\boldsymbol{L}_i)$. Then, we can find the maximum value of $Ent(\bar{\boldsymbol{P}}_i)$ as follows:

$$\arg\max_{P_q} Ent(\bar{\boldsymbol{P}}_i) = \arg\max_{P_q} -\sum_{q=1}^{Q-1}\left[P_q * \log P_q\right]$$

$$s.t. \sum_{q=1}^{Q-1} P_q = 1 - P(\hat{y}_i|\boldsymbol{L}_i). \tag{17}$$

Then, according to the Lagrange multiplier, the Lagrange function $L(\bar{\boldsymbol{P}}_i)$ can be constructed as follows:

$$L(\bar{\boldsymbol{P}}_i) = -\sum_{q=1}^{Q-1}\left[P_q * \log P_q\right] + \lambda(\sum_{q=1}^{Q-1} P_q - 1 + P(\hat{y}_i|\boldsymbol{L}_i)). \tag{18}$$

Now, to obtain the maximum value of $L(\bar{\boldsymbol{P}}_i)$, we can take the partial derivative of $L(\bar{\boldsymbol{P}}_i)$ concerning $P_q$, and set this derivative equal to zero as follows:

$$\frac{\partial L(\bar{\boldsymbol{P}}_i)}{\partial P_q} = -(\log P_q + 1) + \lambda = 0. \tag{19}$$

According to Eq. (19), we can obtain $P_q = 2^{\lambda-1}$. Bringing this result into the constraints, we can obtain $\sum_{q=1}^{Q-1} P_q = (Q-1)2^{\lambda-1} = 1-P(\hat{y}_i|\boldsymbol{L}_i)$. Therefore, it is clear that $P_q = 2^{\lambda-1} = \frac{1-P(\hat{y}_i|\boldsymbol{L}_i)}{Q-1}$. Finally, bringing $P_q = \frac{1-P(\hat{y}_i|\boldsymbol{L}_i)}{Q-1}$ into Eq. (17), we can calculate the maximum value of $Ent(\bar{\boldsymbol{P}}_i)$ as $(1 - P(\hat{y}_i|\boldsymbol{L}_i)) \log \frac{Q-1}{1-P(\hat{y}_i|\boldsymbol{L}_i)}$.

## Appendix B  Derivation of Eqs. (10) - (11)

Due to the limited pages, Eqs. (10) - (12) in the main text only give the derived results. Here, we provide their detailed derivation. First, when $f$ is adjusted, $P(c_q|f, \boldsymbol{x}_i)$ changes with $f$, and this change is denoted as $\Delta_{iq}$. Then, we bring $\Delta_{iq}$ into Eq. (4) and Eq. (5), respectively. The following derivation can be obtained:

$$\tilde{bias}_i^2 = \frac{1}{2}\sum_{q=1}^{Q}\left[P(c_q|\boldsymbol{x}_i) - (P(c_q|f, \boldsymbol{x}_i) + \Delta_{iq})\right]^2$$

$$= \frac{1}{2}\sum_{q=1}^{Q}\left[P(c_q|\boldsymbol{x}_i)^2 + (P(c_q|f, \boldsymbol{x}_i) + \Delta_{iq})^2 - 2P(c_q|\boldsymbol{x}_i)(P(c_q|f, \boldsymbol{x}_i) + \Delta_{iq})\right]$$

$$= \frac{1}{2}\sum_{q=1}^{Q}\left[P(c_q|\boldsymbol{x}_i)^2 + P(c_q|f, \boldsymbol{x}_i)^2 + \Delta_{iq}^2 + 2\Delta_{iq}P(c_q|f, \boldsymbol{x}_i) - 2P(c_q|\boldsymbol{x}_i)P(c_q|f, \boldsymbol{x}_i) - 2\Delta_{iq}P(c_q|\boldsymbol{x}_i)\right]$$

$$= \frac{1}{2}\sum_{q=1}^{Q}\left[P(c_q|\boldsymbol{x}_i)^2 + P(c_q|f, \boldsymbol{x}_i)^2 - 2P(c_q|\boldsymbol{x}_i)P(c_q|f, \boldsymbol{x}_i)\right] + \frac{1}{2}\sum_{q=1}^{Q}\left[\Delta_{iq}^2 + 2\Delta_{iq}P(c_q|f, \boldsymbol{x}_i) - 2\Delta_{iq}P(c_q|\boldsymbol{x}_i)\right]$$

$$= \frac{1}{2}\sum_{q=1}^{Q}\left[P(c_q|\boldsymbol{x}_i) - P(c_q|f, \boldsymbol{x}_i)\right]^2 + \frac{1}{2}\sum_{q=1}^{Q}\left[\Delta_{iq}^2 + 2\Delta_{iq}P(c_q|f, \boldsymbol{x}_i) - 2\Delta_{iq}P(c_q|\boldsymbol{x}_i)\right]$$

$$= bias_i^2 + \sum_{q=1}^{Q}\Delta_{iq}P(c_q|f, \boldsymbol{x}_i) + \frac{1}{2}\sum_{q=1}^{Q}\Delta_{iq}^2 - \sum_{q=1}^{Q}\Delta_{iq}P(c_q|\boldsymbol{x}_i). \tag{20}$$

$$var_i = \frac{1}{2}\Big[1 - \sum_{q=1}^{Q}\big(P(c_q|f,\boldsymbol{x}_i) + \Delta_{iq}\big)^2\Big]$$

$$= \frac{1}{2}\Big[1 - \sum_{q=1}^{Q}\big(P(c_q|f,\boldsymbol{x}_i)^2 + \Delta_{iq}^2 + 2\Delta_{iq}P(c_q|f,\boldsymbol{x}_i))\big)\Big]$$

$$= \frac{1}{2}\Big[1 - \sum_{q=1}^{Q}P(c_q|f,\boldsymbol{x}_i)^2\Big] - \frac{1}{2}\sum_{q=1}^{Q}\Big[\Delta_{iq}^2 + 2\Delta_{iq}P(c_q|f,\boldsymbol{x}_i)\Big] \tag{21}$$

$$= var_i - \sum_{q=1}^{Q}\Delta_{iq}P(c_q|f,\boldsymbol{x}_i) - \frac{1}{2}\sum_{q=1}^{Q}\Delta_{iq}^2.$$

