# OpenReview forum: "IWBVT: Instance Weighting-based Bias-Variance Trade-off for Crowdsourcing"
_NeurIPS.cc/2024/Conference — NeurIPS 2024 poster_

### Official Review · Reviewer_5XV6 · 2024-07-08

**Soundness:** 3
**Presentation:** 2
**Contribution:** 3
**Rating:** 7
**Confidence:** 4

**Summary:**

This paper explores methods to enhance the training of machine learning models using crowdsourced datasets and proposes a novel post-processing approach called IWBVT. The proposed IWBVT first performs instance weighting based on the entropy of the complementary label distribution. Subsequently, it employs a bias-variance trade-off to minimize the generalization error of trained models. Extensive experiment results demonstrate that IWBVT significantly improves the model quality of existing state-of-the-art label integration and noise correction algorithms.

**Strengths:**

1.	This paper proposes a novel post-processing approach that significantly improves the quality of machine learning models trained with crowdsourced datasets. The paper is technically sound, providing detailed derivations and justifications for the proposed approach. The proofs for theorems are clear and rigorous.
2.	The paper is well-written, logical and easy to understand. It clearly describes the problem to be solved and provides a logically rigorous argumentation process. The figures and tables provided in the paper are helpful in illustrating the concept and effectiveness of IWBVT.
3.	The experiments are well-designed, with appropriate datasets and evaluation metrics used to validate the proposed approach.

**Weaknesses:**

1. More detailed explanations about experimental results are desired. For example, in Figure 3b, instance weighting significantly improves model quality on the Leaves dataset, whereas the bias-variance trade-off is more effective on the Income dataset. Understanding the properties of the datasets that lead to these differences would help clarify the conditions under which the proposed approach is most effective.
2. The experiments in this paper primarily focus on the robustness of IWBVT across various label integration and noise correction algorithms and different datasets. To further validate the effectiveness of IWBVT, it would be beneficial to observe its performance on a range of different models.
3. More limitations of IWBVT should be discussed. These include its robustness in the presence of extremely noisy labeling, the computational complexity when handling very large datasets, and potential challenges in real-world applications.

**Questions:**

1.	According to Figure 3b, when improving model quality, why is instance weighting more effective on the Leaves dataset while the bias-variance trade-off is more effective on the Income dataset?
2.	How does IWBVT perform on different machine learning models? Demonstrating that IWBVT can improve the quality of various machine learning models would further validate its effectiveness.
3.	How does IWBVT perform in more complex crowdsourcing scenarios, such as those with extremely noisy labeling or very large datasets? Discussing the potential limitations of IWBVT in these contexts can help enhance its overall effectiveness and applicability.

**Limitations:**

Please refer to the Weaknesses.

---

> ### Author Rebuttal · Authors · 2024-08-05
>
> **Reviewer 5XV6：**
>
> **Q1:** According to Figure 3b, when improving model quality, why is instance weighting more effective on the Leaves dataset while the bias-variance trade-off is more effective on the Income dataset?
>
> **Author Response:** Thanks for your valuable comments. Indeed, in Figure Figure 3b, the instance weighting is more effective on the Leaves dataset, while the bias-variance trade-off is more effective on the Income dataset. This is because the average label quality of the Leaves dataset is low. The instance weighting helps to identify the small number of instances that are correctly inferred and therefore has a greater impact on the Leaves dataset. However, the average label quality of the Income dataset is high, and more instances can be correctly inferred than in the Leaves dataset. Therefore, the bias and variance are estimated closer to the unknown true values, so the bias-variance trade-off is more effective on the Income dataset. In the final version of the paper, we will provide a more detailed explanation of our experimental results. Thanks again for your valuable comments.
>
> **Q2:** How does IWBVT perform on different machine learning models? Demonstrating that IWBVT can improve the quality of various machine learning models would further validate its effectiveness.
>
> **Author Response:** Thanks for your valuable comments. Indeed, we currently used only a simple linear regression (LR) in the probabilistic loss regressions of IWBVT and validated its effectiveness only on Naive Bayes (NB). In fact, IWBVT is effective for other various machine learning models as well. We constructed experiments on the Leaves dataset to validate this conclusion. Specifically, we considered both LR and model tree (MT) as the regression models and used NB and C4.5 as the target models. the experimental results are as follows:
> | |MV| IWMV| LAWMV| MNLDP| AVNC| MVNC| NWVNC|
> |--|--|--|--|--|--|--|--|
> |LR+NB|(59.75, 62.28) ✔|(61.30, 62.85) ✔|(58.74, 60.09) ✔|(55.81, 61.54) ✔|(60.44, 63.02) ✔|(58.44, 60.55) ✔|(58.57, 59.49) ✔|
> |LR+C4.5|(52.10, 55.42) ✔|(52.29, 56.17) ✔|(56.15, 56.64) ✔|(52.69, 58.09) ✔|(56.89, 56.31) ✖|(54.92, 56.31) ✔|(56.12, 58.05) ✔|
> |MT+NB|(59.21, 62.05) ✔|(60.65, 61.45) ✔|(59.21, 62.26) ✔|(55.88, 61.36) ✔|(58.55, 62.15) ✔|(57.97, 59.45) ✔|(61.02, 63.01) ✔|
> |MT+C4.5|(51.96, 55.56) ✔|(51.41, 56.64) ✔|(55.11, 57.06) ✔|(52.09, 56.83) ✔|(57.29, 56.78) ✖|(55.05, 56.53) ✔|(56.74, 58.03) ✔|
>
> Here, "✔" indicates that IWBVT improves the model quality of the corresponding label integration algorithm, while "✖" indicates the opposite. From these results, it can be seen that IWBVT is effective for various machine learning models. In the final version of the paper, we will include a discussion on the effectiveness of IWBVT across different machine learning models. Thanks again for your valuable comments.
>
> **Q3:** How does IWBVT perform in more complex crowdsourcing scenarios, such as those with extremely noisy labeling or very large datasets? Discussing the potential limitations of IWBVT in these contexts can help enhance its overall effectiveness and applicability.
>
> **Author Response:** Thanks for your valuable comments. Our proposed IWBVT is not restricted to specific crowdsourcing scenarios, so we conducted experiments on the whole 34 simulated datasets published by the CEKA platform. These datasets contain some large datasets such as the letter dataset. The experimental results in Table 1 indicate that IWBVT is effective on these large datasets. Additionally, the real-world dataset Leaves, whose percentage of noisy labels exceeds 0.4, is an extremely noisy labeling dataset. The experimental results shown in Figure 2 indicate that IWBVT is effective on this extremely noisy labeling dataset as well. In Section 6, we have discussed some limitations of IWBVT. In the final version of the paper, we will further refine our explanation for these limitations. Thanks again for your valuable comments.

---

> > ### Comment · Reviewer_5XV6 · 2024-08-12
> >
> > I am satisifed with the authors' rebuttal and keep on original scoring and confidence.

---

### Official Review · Reviewer_fq5P · 2024-07-11

**Soundness:** 4
**Presentation:** 3
**Contribution:** 3
**Rating:** 7
**Confidence:** 4

**Summary:**

This paper proposes a novel label integration method called IWBVT for crowdsourcing by using a weighting method and probabilistic loss regressions to improve the model quality. The main problem this paper solves is the model quality caused by the presence of the intractable instances. The paper is well organized and the writing is smooth. The method proposed in the paper has improved the model quality, thus boosting the performance on simulation and real-world datasets. The authors also provide theoretical proof to deepen the understanding of the proposed method.

**Strengths:**

1. The paper is well organized and the writing is smooth. The authors provide necessary symbol explanations, which is very helpful for reading.
2. The idea proposed in this paper is technical soundness. The authors incorporate instance weighting and bias-variance trade-off components to complete the overall algorithm. Furthermore, the authors also provide the corresponding theory to prove the effectiveness and generalization of the designed components. Experiments on simulation and real-world datasets demonstrate the superiority of the proposed method.
3. For instance weighting component, the authors prove that existing methods [1] and [2] are special cases of the method proposed by the authors. Hence, the method proposed by the author demonstrates strong generalization and can be applied to a wider range of situations.

[1]. V.S. Sheng, F.J. Provost, and P.G. Ipeirotis. Get another label? Improving data quality and data mining using multiple, noise labels. SIGKDD, 2008.

[2]. Z. Chen, L. Jiang, and C. Li. Label augmented and weighted majority voting for crowdsourcing. Inf. Sci, 2022.

**Weaknesses:**

1. Compared with real-world dataset, experimental results on a few simulated datasets doesn’t improve (e.g., MV for breast-cancer/breast-w datasets). Furthermore, the number of loss for MV and IWMV is more than that for other methods in Table 1. I suggest adding some discussion about this phenomenon.
2. Main experiments adopt significance level to indicate the performance for each method. How to decide the value of alpha and what is the influence? More detail will be helpful.

**Questions:**

Please refer to weakness.
Other questions:
1. In the experimental section, why are methods like significance testing used instead of direct comparison?

**Limitations:**

yes.

---

> ### Author Rebuttal · Authors · 2024-08-05
>
> **Reviewer fq5P：**
>
> **Q1:** Compared with real-world dataset, experimental results on a few simulated datasets doesn’t improve (e.g., MV for breast-cancer/breast-w datasets). Furthermore, the number of loss for MV and IWMV is more than that for other methods in Table 1. I suggest adding some discussion about this phenomenon.
>
> **Author Response:** Thanks for your valuable comments. Our proposed IWBVT is not restricted to specific crowdsourcing scenarios, so we conducted experiments on the whole 34 simulated datasets published by the CEKA platform. However, no algorithm can achieve the best performance on all datasets. Indeed, IWBVT does not improve MV on a few datasets, such as breast-cancer and breast-w. This is because the performance of IWBVT is affected by the performance of the label integration algorithm it aims to improve. According to Eqs. (6) and (9), IWBVT depends on integrated labels inferred by label integration algorithms for its instance weighting and probabilistic loss regressions. Therefore, noise in these integrated labels affects the performance of IWBVT. MV and IWMV, as classical label integration algorithms, generally perform worse than other state-of-the-art algorithms, resulting in integrated labels inferred by MV and IWMV containing more noise. This ultimately leads to more losses for MV and IWMV compared to other state-of-the-art algorithms in Table 1. In the final version of the paper, we will include these discussions to explain this phenomenon. Thanks again for your valuable comments.
>
> **Q2:** Main experiments adopt significance level to indicate the performance for each method. How to decide the value of alpha and what is the influence? More detail will be helpful.
>
> **Author Response:** Thanks for your valuable comments. The significance level is a crucial concept in statistics that directly affects the results of the corrected paired two-tailed t-test used in our experiment. In a t-test, the significance level is typically set to a fixed value, such as 0.05 or 0.1. If the p-value is less than this significance level, we reject the null hypothesis, indicating a significant difference between the two comparison algorithms. Therefore, the significance level affects the stringency of the test: a lower significance level (e.g. 0.05) implies a more stringent test, as the null hypothesis is rejected only when the evidence is very strong. Conversely, a higher significance level (e.g. 0.1) is relatively relaxed and more likely to accept the null hypothesis. In the final version of the paper, we will include a more detailed explanation of the significance level. Thanks again for your valuable comments.
>
> **Q3:** In the experimental section, why are methods like significance testing used instead of direct comparison?
>
> **Author Response:** Thanks for your valuable comments. To exclude the effect of randomness, our simulated experiments were independently repeated ten times. In each experiment, we evaluated the model quality using 10-fold cross-validation. Therefore, each label integration algorithm produced 100 pairs of comparison results (original model quality vs. IWBVT improved model quality) on each dataset. In Table 1, we reported the averages (arithmetic mean) of these results to provide a general indication of relative performance. However, these averages do not indicate whether the comparison results are significantly different. Therefore, to more accurately evaluate the performance of IWBVT, we statistically compared these results by the corrected paired two-tailed t-test. The corrected paired two-tailed t-test determines whether IWBVT effectively improves the corresponding label integration algorithm by assessing whether the differences between the 100 pairs of comparison results satisfy the null hypothesis. Thanks again for your valuable comments.

---

> > ### Comment · Reviewer_fq5P · 2024-08-12
> > **Response for Rebuttals.**
> >
> > Thank you for the helpful response that addressed my concern.

---

### Official Review · Reviewer_Td4g · 2024-07-13

**Soundness:** 3
**Presentation:** 2
**Contribution:** 3
**Rating:** 6
**Confidence:** 4

**Summary:**

The paper studies the problem of improving quality of models trained on datasets collected through crowdsourcing. The authors propose an approach (IWBVT) that post-processes data after crowdsourcing with the goal to mitigate the impact of intractable instances by means of instance weighting. As a result, the bias and variance of trained models becomes closer to the unknown true labels. It is proven that the novel method reduces the generalization error of trained models by the bias-variance trade-off. The paper also contains extensive experimentation that demonstrates: IWBVT significantly improve the model quality of existing label integration algorithms and noise correction algorithms.

**Strengths:**

-	Novel method

-	Extensive experimentation over 34 simulated datasets and 2 real ones

**Weaknesses:**

-	Formalization of results

-	Clearness of explanation in some points behind the novel approach

(see Questions)

**Questions:**

- I do not understand the message behind Figure 1. How should I tract the 4 distributions on the right side of the arrow?

- Lines 164 – 165: “it can be verified that Eq. (6) can distinguish all complex distributions we showcased in Section 3.1. By..” It is hard to assess the value of this claim, because it is not clear how broad complex distributions are covered. Is it just 2 among 100? Among 1000 possible? Any quantifiable / qualifiable ways to measure the distinguish power for complex distributions?

- Line 166 “Theorem 1. In some special cases,” Please, specify cases in the theorem. Otherwise, I would not agree that this might be a theorem statement. The definition of the cases must be contained outside of the proof. Right now, I can treat the theorem as always true, because “some special cases” might be = \emptyset .

- Similar, Line 185 in Theorem 2: “Eq. (8) helps Eq. (7)”… The word “help” does not have formal mathematical definition, while Theorem is a mathematical instrument.

**Limitations:**

the authors adequately addressed the limitations

---

> ### Author Rebuttal · Authors · 2024-08-05
>
> **Reviewer Td4g：**
>
> **Q1:** I do not understand the message behind Figure 1. How should I tract the 4 distributions on the right side of the arrow?
>
> **Author Response:** Thanks for your valuable comments. The four distributions on the right side of the arrow in Figure 1 correspond to the four cases of our new instance weighting method. Among them, the distribution in the upper left corner indicates that when the entropy of the complement $Ent(\bar{P}_i)$ is fixed, a lower $P(\hat{y}_i|L_i)$ (the probability of the integrated label in multiple noisy label distribution) results in a lower weight $w_i$ assigned to the instance. Conversely, the distribution in the upper right corner indicates that a higher $P(\hat{y}_i|L_i)$ results in a higher $w_i$. The distribution in the lower left corner indicates that when $P(\hat{y}_i|L_i)$ is fixed, a lower $Ent(\bar{P}_i)$ results in a lower $w_i$. Conversely, the distribution in the lower right corner indicates that the higher $Ent(\bar{P}_i)$ results in a higher $w_i$. These explanations are provided in lines 154-162, and we will refine them further in the final version of the paper. Thanks again for your valuable comments.
>
> **Q2:** Lines 164 – 165: “it can be verified that Eq. (6) can distinguish all complex distributions we showcased in Section 3.1. By..” It is hard to assess the value of this claim, because it is not clear how broad complex distributions are covered. Is it just 2 among 100? Among 1000 possible? Any quantifiable / qualifiable ways to measure the distinguish power for complex distributions?
>
> **Author Response:** Thanks for your valuable comments. To further demonstrate the distinguish power of Eq. (6) for complex distributions, we employed a quantifiable way. Specifically, on all the complex distributions exemplified in Section 3.1, the mentioned instance weighting methods produced the following results:
> |Complex distributions|A|B|C|D|
> |--|--|--|--|--|
> |{0.5, 0.3, 0.2} and {0.5, 0.4, 0.1} (on line 122)|(0.50, 0.50) ✖|(0.67, 0.73) ✖|(0.20, 0.10) ✔|(0.70, 0.52) ✔|
> |{0.4, 0.3, 0.3} and {0.4, 0.4, 0.2} (on line 127)|(0.40, 0.40) ✖|(0.64, 0.66) ✖|(0.10, 0.00) ✔|(0.58, 0.53) ✔|
> |{0.5, 0.3, 0.1, 0.1} and {0.4, 0.2, 0.2, 0.2} (on line 131)|(0.50, 0.40) ✔|(0.59, 0.52) ✔|(0.20, 0.20) ✖|(0.62, 0.58) ✔|
>
> Here, A, B, C, and D denote $P(\hat{y}_i|L_i)$, $\frac{1}{Ent(P_i)}$, $max(P_i)-sec(P_i)$ and our new instance weighting method, respectively. "✔" and "✖" indicate whether the weighting method is effective in distinguishing the corresponding complex distribution, respectively. Empirically, the weight corresponding to the front distribution in these examples should be greater than the latter. The results show that our new instance weighting method can distinguish all types of complex distributions that existing methods cannot. In the final version of the paper, we will include these comparisons and analyses to demonstrate the distinguish power of our new method. Thanks again for your valuable comments.
>
> **Q3:** Line 166 “Theorem 1. In some special cases,” Please, specify cases in the theorem. Otherwise, I would not agree that this might be a theorem statement. The definition of the cases must be contained outside of the proof. Right now, I can treat the theorem as always true, because “some special cases” might be = \emptyset .
>
> **Author Response:** Thanks for your valuable comments. Indeed, the special cases mentioned in Theorem 1 should be specified. In the final version of the paper, we will move the definition of these special cases from the proof of Theorem 1 to Theorem 1 itself. Specifically, the refined Theorem 1 will be "When $Ent(\bar{P}_i)$ remains constant, Eq. (6) covers $w_i \propto P(\hat{y}_i|L_i)$. When $Q>2$ and $P(\hat{y}_i|L_i)$ is the maximum value in $P_i$, Eq. (6) covers $w_i \propto \max(P_i) - \sec(P_i)$.". Thanks again for your valuable comments.
>
> **Q4:** Similar, Line 185 in Theorem 2: “Eq. (8) helps Eq. (7)”… The word “help” does not have formal mathematical definition, while Theorem is a mathematical instrument.
>
> **Author Response:** Thanks for your valuable comments. Indeed, we should use a more formal mathematical definition in our Theorem 2. In the final version of the paper, Theorem 2 will be refined as follows: "When the probabilistic loss is defined as in Eq. (9), performing probabilistic loss regressions constructed by Eq. (8) ensures that Eq. (7) asymptotically achieves the bias-variance trade-off.". Thanks again for your valuable comments.

---

> ### Author Response · Authors · 2024-08-12
>
> As the discussion period deadline nears, we would be deeply appreciative if you could kindly review our rebuttal and let us know if we have addressed your concerns. We’re more than happy to continue the conversation if you have any further questions. Thank you very much for your time and consideration.

---

### Decision · Program_Chairs · 2024-09-25

**Decision:**

Accept (poster)

**Comment:**

This paper proposes a new instance weighting-based approach to improving the quality of models trained on datasets collected through crowdsourcing, and theoretically demonstrates that an appropriate bias-variance trade-off is achieved.
The paper makes a significant technical contribution to a major problem in this topic, and the issues raised by the reviewers have been appropriately addressed, and further enhanced the completeness of this work.